# A Thermal Model for Processing Data from Undergarment Sensors in Automatic Control of Actively Heated Clothing

Wojciech Tylman [1,*], Rafał Kotas [1], Marek Kamiński [1], Sebastian Woźniak [1] and Anna Dąbrowska [2]

1    Department of Microelectronics and Computer Science, Lodz University of Technology, 93-005 Lodz, Poland; rafal.kotas@p.lodz.pl (R.K.); marek.kaminski@p.lodz.pl (M.K.); sebastian.wozniak@dokt.p.lodz.pl (S.W.)
2    Department of Personal Protective Equipment, Central Institute for Labour Protection—National Research Institute, 90-133 Lodz, Poland; andab@ciop.lodz.pl
*    Correspondence: wojciech.tylman@p.lodz.pl

**Abstract:** Despite its recent growth in popularity, actively heated clothing still lacks the ability to cope with demanding user scenarios. As many of these deficiencies stem from an absence of automatic control, the authors propose a novel approach using a set of sensors embedded in the clothing to provide data about thermal comfort. Available sensors suffer from a lack of accuracy, as for practical reasons, they cannot be attached to the skin, whose temperature is usually used as a comfort indicator. To determine the magnitude of the problem, the authors conducted experiments, and a thermal model was proposed based on experimental findings; the output from the model was compared with the experimental reference data for three different upper body undergarments. The overall accuracy was found to be good: in most cases, the difference between the computed and reference skin temperatures did not exceed 0.5 °C. Furthermore, the model does not rely on unrealistic assumptions regarding the availability of parameters or measurement data. Our findings demonstrate that it is possible to create a thermal model that, when used for input data processing, allows undergarment temperature to be converted to skin temperature, allowing for automatic control of heating insets.

**Keywords:** actively heated clothing; automatic control; embedded software; thermal comfort; thermal model

## 1. Introduction

### 1.1. Motivation

The work of mountain rescuers is very often performed in difficult conditions. As these conditions commonly require long hours of activity at low temperatures and high humidity, the rescuers would profit from personalised and specialised thermally active protective clothing rather than the passive, general-use hiking clothing that most currently use.

The obvious solution is to heat the clothing to improve the user's thermal comfort. However, as described in Section 1.3, the available solutions do not meet all of the requirements of mountain rescuers. The three most frequently indicated shortcomings comprise insufficient battery autonomy associated with the limited capacity of the battery and the lack of advanced power management, low heating power and the possibility for only basic, manual, on/off controls. However, thanks to modern technology, it is possible to design and implement systems that will address the abovementioned problems.

The authors of the present article are currently developing a "Personalized Protective Thermally Active clothiNg (sPParTAN)" system. It consists of an electronic part (embedded microprocessor system, textile heaters, intelligent power supply circuit) combined with a comprehensive IT platform (mobile application, database) that allows for full control and monitoring of the intelligent clothing.

### 1.2. Aim

The overall goal of the developed system is to maintain the thermal comfort of the user. To achieve this goal, heating insets are controlled by an automatic algorithm based on input data from sensors. Based on previous studies (see the discussion of how to determine thermal comfort, in Section 1.3), skin temperature was selected as the main input quantity; as such, there is a need for a sufficiently precise means of measuring this temperature. On the other hand, as the user has to be able to don the garments quickly, the temperature sensors have to be integrated within the garment containing the heating insets. Consequently, as they cannot be attached to the skin, the sensors measure the temperature of the undergarment climate rather than the skin temperature.

Therefore, the aim of the present work was to confirm whether the temperature of the undergarment climate can be directly used for controlling the heaters instead of the skin temperature. If this is not possible, then the study should:

- Investigate the magnitude of errors introduced by the proposed setup;
- Determine whether the type of underclothing has a significant influence on these errors; and
- Establish a reliable thermal model that allows the automatic algorithm to correctly use the temperature data from undergarment sensors.

Regarding the last aim, the model must consider the potential variability in undergarments (the outer garment being a fixed element of the clothing set), yet it must not rely on unrealistic assumptions regarding the availability of parameters or measurement data. This means that the required parameters should be tied only to the type of undergarment used, and not to any characteristics of the user or the environment; in addition, the only measurement data should be those readily measurable during the work of a typical mountain rescuer; ideally, the data should be obtained from the sensors already present in the system.

### 1.3. State-of-the-Art

Maintaining comfortably high body temperature has been a challenge for *Homo sapiens* since his migration out of Africa some 215,000 years ago. Clothing was arguably invented primarily for this very purpose; until the last century, it was available only in its passive form, with the body acting as the sole source of heat and the clothing merely a passive barrier to isolate it from the outer environment. The ubiquity of this approach obfuscates its numerous disadvantages, one of the most critical being the dependence of the thermal resistance of clothing on its thickness; despite recent developments in fabric and filling technology, clothes intended for extremely cold weather or prolonged exposure to cold tend to be thick. This thickness in turn impairs freedom of movement. Another issue is that a given set of clothes has a constant thermal resistance, and therefore cannot adapt to changes in internal (heat production through bodily exertion) or external (temperature, wind, insolation) conditions.

The first steps towards actively heated clothes were those associated with the development of electric blankets (or heating pads) at the beginning of the 20th century [1,2]. Heating jackets were available for military plane crews in the 1940s [3], and patents for actively heated clothes were filed [4]. Research in methods to integrate heating elements within fabrics still continues, and solutions utilising silver, copper or carbon nanotubes have been tested [5]. The fabrics themselves have also undergone significant evolution towards adaptive materials that can efficiently manage heat transfer between the human body and the environment [5]. For example, textiles utilising fibre-based artificial muscles may react to sweating through contraction, thus exposing a larger skin area for improved cooling [6]. Combining different types of fabrics can also significantly affect the overall performance of clothing, allowing for clothes especially suited for a particular activity or climate [7]. Using modern technology, smart garments can also be designed to target very specific needs, such as treatment of physical disorders [8].

As indicated by a review of previous literature [9] and the products available through retail chains [10–12], contemporary commercially available heated clothes usually have

limited heating power, insufficient heated area and inadequate battery autonomy; in addition, a particularly important point in view of this article, they rely on simple manual controls that do not permit any fine adjustment of heating power or allow zones in the garment to be heated differently. Basically, they only have an on/off switch. A clear area for improvement would be to include a set of sensors to provide input for automatic control. While some research has been performed in this field [13], it is not common. The main challenge is obtaining data that reliably describe the level of thermal comfort.

Much research has been devoted to methods of determining whether a person is in thermal comfort, and this is no trivial task. The often used Fanger equation [14] requires no less than eight input values, many of which are difficult to measure, even under laboratory conditions, and the possibility of measurement in a commercial clothing product intended for active outdoor use is out of the question. Although several analytical or numerical thermal models with various levels of complexity exist [15], they require a range of parameters, in addition to input values, which are not available outside the laboratory. Fortunately, some approximations are possible, tying thermal comfort to skin temperature [5,16] and wetness [17] with relation to metabolic heat production. For example, empirical Equation (1) [16] ties the skin temperature required for comfort ($t_{sk}$) with the metabolic heat production ($M$) and the external mechanical work ($W$). Metabolic heat production is in turn estimated for typical activities [15]. Note that the external mechanical work is no more than 25% of the total metabolic rate and 0 for many activities [15].

$$t_{sk} = 35.7 - 0.0275(M - W)°\text{C} \tag{1}$$

However, further research is needed. Firstly, as such a simple estimate may not be valid in all situations and all conditions, a method of adapting the automatic algorithms to the preferences of the user is needed. Secondly, and this is the main topic of the present article, proper skin temperature measurements may not be possible in thermally active clothing. Even with sensors affixed to the skin, substantial differences in readings can be obtained depending on the pressure applied, environmental conditions and whether the back of the sensor is covered or not [18]. In a commercial product that has to be quick to don, comfortable to use and reliable and inexpensive, affixing the sensors to the skin is not possible: thermal sensors are embedded in the clothes and are metrologically imperfect.

Some studies have targeted this type of setup, but they remain inconclusive. One study [19] examines the use of optical sensors embedded in the clothing. Although it proposes a model of heat transfer and the resulting measurement accuracy is good ($\pm 0.18$ °C compared with medical mercury thermometer), the setup is somewhat involved and the authors do not consider a situation when an undergarment is placed between the sensor and the skin. Another study describes a more conventional approach based on the use of an electronic LM35DM sensor [20], with silicone pads being employed to improve heat transfer from skin to the sensor. In this case, the mean measurement error ranged from 0.278 °C to 0.844 °C, depending on the setup. However, these measurements were taken under steady state conditions at ambient room temperature. Again, no provisions for the presence of an undergarment were made.

A third article discusses the concept of integrating thermocouple sensors in the fabric [21]; however, the authors focus on the manufacturing process and do not consider real-life measurement precision. A final study discusses the use of yarn-based sensors woven into textiles [22]; while interesting results regarding the time constants of these sensors are presented, they were obtained by affixing the material to a hot plate, which differs from real-life conditions. Again, no influence of sub-zero external temperatures was investigated.

## 2. Materials and Methods

### 2.1. Overview of the System

Figure 1 presents a block diagram of the system. The system consists of many elements, not all of which were utilised in the described experiments.

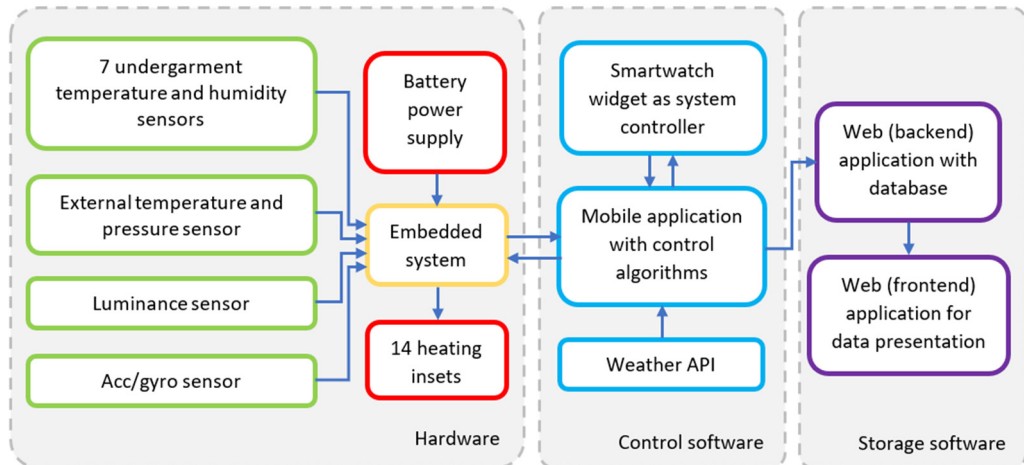

**Figure 1.** Block diagram of the system.

The most important element is the clothing, consisting of a thermal union suit, equipped with heating insets and temperature and humidity sensors. Their placement in the suit is given in Figure 2.

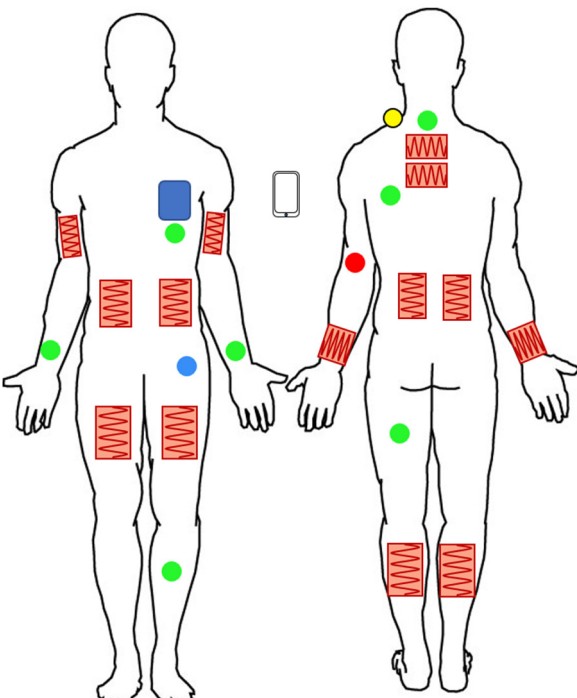

**Figure 2.** Placement of system components in the suit—left front, right back (red rectangles—heating insets; blue rectangle—embedded system; green circles—undergarment temperature sensors; blue circle—acc/gyro sensor; red circle—external temperature sensor; yellow circle—luminance sensor).

As for the sensors, their placement is a compromise between three necessities:

- Adherence to typical placement used for the estimation of mean skin temperature over the body, according to IEC norm ISO 9886 (left palm, right shin, nape, right scapula) [15];
- The requirements of functional convenience indicated by the end users—in particular, some locations must be excluded because they interfere with rescuers' equipment such as backpack, harness, etc.; and
- Ability to gather data about local thermal comfort of different parts of the body.

In the case of insets it was decided to place them both in areas critical from the point of view of human survival (i.e., abdomen) and in places where keeping warm influences the subjective feeling of thermal comfort (i.e., legs and arms).

The insets and sensors are connected to a microprocessor-based embedded system, which is connected in turn to an application running on a smartphone. The application allows the user to set the heating levels of the insets, while gathering data from the embedded system. The application can also connect over Wi-Fi or an LTE connection to the backend web application in order to transmit the collected data using a communication protocol based on the JSON format. The backend application stores data in a database, from which data can subsequently be retrieved using a frontend web application.

### 2.2. Clothing

The objective of the project is to develop a set of personalised clothing consisting of an outer windproof Gore-Tex jacket, a down jacket, windproof trousers and an inner suit with an integrated active heating system. The dimensions of the clothes can be personalised based on anthropometric measurements acquired from digital photos of each rescuer's silhouette. The clothing design and production are the responsibility of one of the partners in the project, PSA Małachowski, highly specialised producers of winter outerwear in Poland.

### 2.3. Sensors and Insets

To design an automatic algorithm for maintaining thermal comfort, a set of sensors measuring temperature, humidity, luminance and pressure, as well as accelerometers/gyroscopes, was implemented in the clothing (Figures 1 and 2). Data from the temperature and humidity sensors are recorded at a frequency of 1 Hz. All sensors were placed inside robust universal housings that were specially adapted, i.e., had holes drilled, to enable air flow or light penetration. The size of the sensor presented in Figure 3 is about 3 cm × 4 cm × 1 cm.

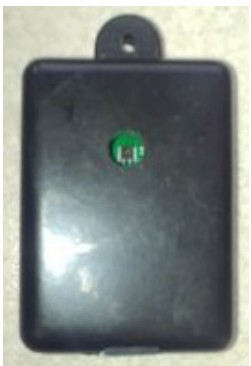

**Figure 3.** Temperature and humidity sensor used in the clothing prototype.

The most important sensors are those measuring temperature and humidity. The external temperature is measured with Bosch BMP280 sensor. It provides both pressure and temperature readings. The undergarment climate is measured with the use of seven Sensirion SHTC3 sensors. This sensor provides both temperature and humidity readings. For all sensors, $I^2C$ communication protocol is employed. Metrological characteristics of the sensors are provided in Table 1.

The construction presented in Figure 3 is the compromise between sensor robustness and tactile comfort of the end user. However it should be noted that the presented work is a part of an ongoing project and therefore tactile comfort was not the primary goal of this experiment. It will be further developed, taking into account the comments and opinions of end users. Moreover, the RC thermal model presented in detail in Section 5 enables change

of the time constant depending on the design of the sensors. Therefore it could be easily adapted to newer, more user-friendly sensor casing.

**Table 1.** Sensors parameter specification.

| Sensor | Measured Parameter | Resolution | Range | Accuracy |
|---|---|---|---|---|
| Bosch BMP280 | External temperature | 0.01 °C | −40 ... +85 °C | ±0.5 °C |
| | Pressure | 0.16 Pa | 300 ... 1100 hPa | ±1 hPa |
| Sensirion SHTC3 | Temperature | 0.01 °C | −40 ... +125 °C | ±0.2 °C |
| | Humidity | 0.01% RH | 0 ... 100% RH | ±2.0% RH |

The sensors and insets are connected to the embedded system using thin and quite flexible wires. They have been routed in tunnels along the seams of the garment to minimise their negative impact on user comfort. The sensors are attached to the clothes with Velcro. The insets are sewn onto the clothes. The tests carried out so far have revealed no problem in this respect.

The heating insets are the effectors of the system, and therefore they are its most crucial part. However, as the heating performance is not analysed in this article, details of their construction will not be included.

### 2.4. Embedded System

The central point of the solution is the embedded system, consisting of two modules: a microprocessor-based control unit and a power supply module. The control module is based on an ARM Cortex-M4 microprocessor with an integrated radio transceiver. It employs I2C as a communication protocol with all sensors and Bluetooth Low Energy to communicate with the mobile application. The power supply module distributes the energy from the battery pack to the heating insets (using pulse width modulation to control heating levels) and all electronic subsystems. One of the assumptions of the system is for it to have operational reliability; hence, it was decided to use wired connections between the embedded system and sensors.

### 2.5. Mobile Application

The mobile application has two tasks: to control the heaters in the clothing and to gather and transmit data. A block diagram of the mobile application is presented in Figure 4, and the GUI in Figure 5. The user can manually control the heating insets with the use of dedicated buttons (red- and blue-coloured buttons in Figure 5) or they can activate one of two automatic algorithms (two green buttons in the lower part of the screen in Figure 5). One of the algorithms also uses artificial neural networks to better adjust the performance to the needs of the user. Pressing small green buttons in each body section opens a dialog allowing the user to indicate the comfort level based on a scale of "too cold", "comfortable" and "too hot". Each time the user switches the heaters on or off or reports that the temperature of the heater is too low or high, this information is stored in the database.

As the application uses cross-platform technology, Capacitor [23], there is no need to rewrite the code for two separate platforms (Android and iOS). It connects with the embedded system via Bluetooth Low Energy, and data are sent to backend applications by the HTTP protocol.

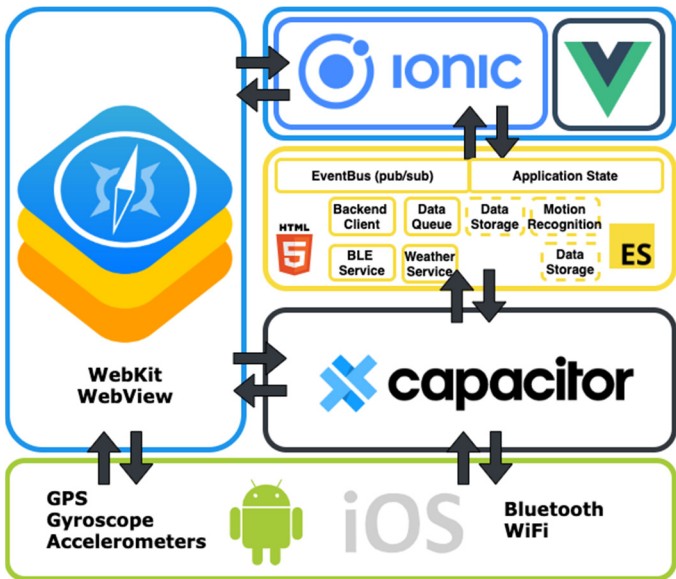

**Figure 4.** Block diagram of the mobile application.

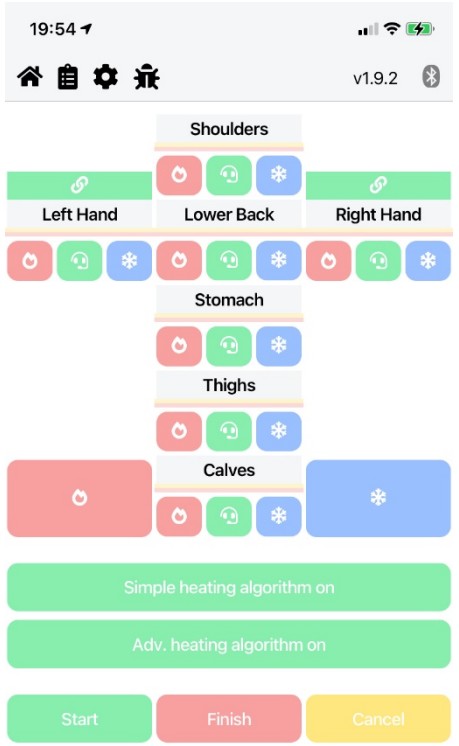

**Figure 5.** Mobile application GUI.

## 2.6. Backend and Frontend Web Applications

This allows one to gather the data from the experiment and therefore it is an important part of the system. In order to allow the application to be continuously adapted to other, regularly changing elements of the project, it was designed as a flexible element then can be easily modified and extended. It allows stable collection of data generated during experiments, and supports the validation and correction of transmitted data. It communicates with the mobile application via the REST API and with users via the frontend application.

The frontend application is a graphic user interface allowing the user to browse, visualise and download data.

### 3. The Experiment

The experiment was conducted in a mobile freezer chamber which maintained an air temperature of approximately $-5\,^\circ$C. The dimensions of the chamber were 4.20 m (length) by 2.15 m (width) by 2.10 m (height). The only source of air movement in the chamber was a group of moderately sized fans connected to the evaporator mounted inside the chamber. The airflow was directed close to the ceiling of the chamber, so the forced air movement around the persons in the chamber was minimal.

The experiment was designed to test long-time transient responses of the complex body-clothing-sensors system, without disturbances introduced by variable metabolic heat production. Consequently, the participants remained stationary at all times.

The temperature and humidity during the experiment were recorded using two sets of sensors: the first were the sensors of the thermally active clothing, as described in Section 2.3. The second set, treated as reference, comprised a group of Maxim Integrated iButton sensors (model DS1923, Figure 6) placed in direct contact with the skin, close to the clothing sensors, and affixed using strips of adhesive tape. As they do not support continuous data transmissions, the recorded measurements had to be read after the conclusion of the experiment. Furthermore, the obtained data had to be synchronised with the clothing sensors using timestamps recorded by both sets of sensors—for this purpose a dedicated Matlab script was written. At first, the signals were resampled by the *retime* function using linear interpolation with 0.25 s step. Then, the *synchronise* method was used to aggregate data from all input timetables and adjust them to common time vector.

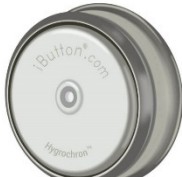

**Figure 6.** Maxim Integrated iButton DS1923 sensor.

Donning of the undergarments (if used), the thermoactive clothing and the outerwear trousers and jacket was performed at room temperature prior to entering the chamber; the clothing was not altered during the experiment.

The participants entered the chamber and remained there for up to 40 min. During the experiment, the participants were asked to report their thermal comfort immediately upon experiencing any change. This was accomplished through the mobile application (see Section 2.5 and Figure 5): the participants indicated one of the seven measured zones, viz., shoulders, lower back, stomach, left hand, right hand, thighs or calves, and selected one of three reports: "comfortably", "too cold" or "too hot". In stage 5 of the experiment, the participants were also allowed to control the heating level of the insets (from 0 to 100%, in 20% steps); however, in the remainder of the stages, the insets were turned off or set to maximum power. The selected heating level was recorded in the experiment data.

The scenario for the experiment is presented in Table 2. Most of the presented analyses apply to stages 2–3; although in most cases, the observed influence of the heating insets on the measured temperatures was minimal, more reliable analyses can be performed working with data with no known disturbances. For all the participants, these stages lasted at least 1000 s.

**Table 2.** Sequence of events of the experiment.

| Stage | Task |
|---|---|
| 1 | Don the clothes, including the Gore-Tex jacket. |
| 2 | Enter the chamber. |
| 3 | Stand still until "too cold" sensation is experienced in all zones or the temperature sensation becomes stable. |
| 4 | Turn on heating insets. |
| 5 | Stand still and adjust the heating level until a "comfortable" sensation is experienced in all zones or, if this cannot be attained, the temperature sensation becomes stable. |
| 6 | Set the heating level to maximum. |
| 7 | Stand still until "too hot" sensation is experienced in all zones, or the temperature sensation becomes stable or the heat sensation is significantly unpleasant in any zone. |
| 8 | Turn off the heating insets. |
| 9 | Stand still until a "comfortable" or "too cold" sensation is experienced in all zones or the temperature sensation becomes stable. |
| 10 | Exit the chamber. |

## 4. Results

The following terminology and conventions are used consistently across this section:

- Reference sensor—the iButton skin temperature and humidity sensor;
- Clothing sensor—the sPParTAN project undergarment temperature and humidity sensor;
- External sensor—the sPParTAN project external temperature and pressure sensor.

In the experiment, three male participants took part. Their characteristics are described in Table 3.

**Table 3.** Characteristics of participants.

| Participant | Age | Height | Weight |
|---|---|---|---|
| A | 36 | 188 cm | 85 kg |
| B | 40 | 172 cm | 65 kg |
| C | 46 | 179 cm | 76 kg |

To determine how the transient responses of the complex body-clothing-sensors system were affected by the clothing, three different approaches to the upper body undergarment were used by the participants: A—no underwear, B—thin sweat-wicking t-shirt, C—moderately thick long-sleeve thermal underwear.

The results for the three participants' shoulder sensors (clothing and reference) and external sensors are presented in Figure 7. The points of interest are:

- A significant discrepancy was observed between the temperatures reported by the clothing and reference sensors. This discrepancy varies in time and differed between participants A, B and C.
- The temperature readouts for the instant in time when the participants first reported "too cold" vary between participants, as indicated by both the clothing and the reference sensors.
- It takes approximately 40 min from entering the chamber for the skin temperature to reach a steady state, as indicated by the reference sensor of participant A. Note that the almost flat response for participant B, starting from minute 18, is not due to reaching

the steady state, but is due to some rise in ambient temperature influencing the sensor response.

- The external temperature sensor responds to air temperature changes with a significant delay. In an ideal case, its reading should drop to the air temperature of the chamber (−5 °C) immediately upon entrance of the participant into the chamber; instead, it takes between 9 and 10 min to reach a readout of 0 °C. Note that for participant A, the external temperature curve is not smooth and the time to reach 0 °C is 20 min—this may indicate that the air temperature in the chamber was not uniform and the participant moved around the chamber.

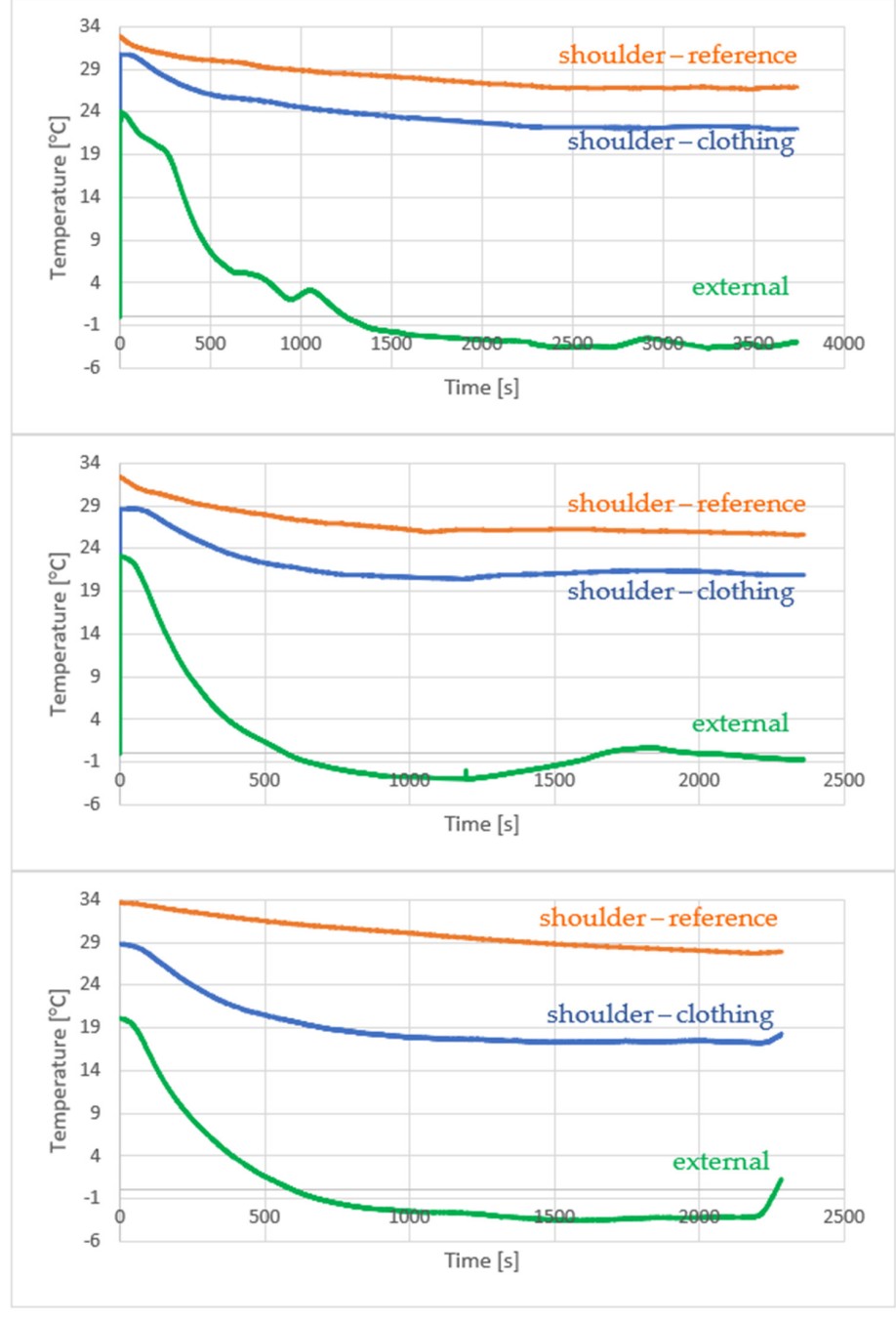

**Figure 7.** Temperature sensor readings: shoulder—reference (orange), shoulder—clothing (blue) and external (green) temperatures. From top: participants A, B and C. Note the different duration for participant A. The sudden rise of the external temperature reading for participant C at 2200 s is caused by leaving the chamber.

## 5. Discussion

The obtained results clearly show that the clothing sensors provide different information than the reference sensors. For a system that should automatically maintain thermal comfort, this is a significant problem.

Skin temperature may be viewed as a valid indicator of thermal comfort [14,24]. Although it is not the best estimator in this regard, it is impossible to use more accurate methods in the proposed system, as they require measurements impossible to reliably and repetitively perform on persons performing their duties as mountain rescuers [25]. As the range of skin temperatures over which thermal comfort is experienced is relatively narrow [26], it is important to eliminate any possible observed discrepancies to provide accurate heating control.

The following analyses and computations apply to the first 1000 s after entering the freezing chamber. After that time, some of the participants turned on the heating insets; although this did not introduce significant changes in the temperature readings, the data obtained from this point onward are less reliable. The phenomena occurring after this point will be subject to individual analysis in a separate article.

### 5.1. Effect of In-Clothes Sensor Placement

After analysing the obtained data, the authors propose that the main reason for the observed discrepancies between the clothing and reference sensors, and the differences in severity between participants, is the fact that the temperature of the undergarment microclimate, which the clothing sensor is measuring, is heavily influenced by the type of undergarment. Verily, assuming that the external temperature is not equal to the skin temperature, the temperature of the undergarment microclimate cannot be identical to that of the skin, due to the temperature gradient over the clothing and air layers. As in the experiment the outer clothing was identical for all the participants, and the clothing and reference sensors were carefully placed close to each other, the only factor significantly different between the participants remains the underwear.

In order to circumvent this problem, a thermal resistance model has been proposed (Figure 8). Skin temperature ($T_s$) can be computed from the clothing sensor and external sensor temperatures ($T_m$ and $T_e$) according to Equation (2). The value of $R_x$ can be eliminated from the equation, as only the ratio of two thermal resistances is of importance (an example of a thermal divider). The parameter $a$ can be computed by requiring the computed and measured (by the reference sensor) skin temperatures to be equal. Before actual computations, it is necessary to account for another problem, as discussed in the next section.

$$T_s = T_m + a \cdot R_x \frac{T_m - T_e}{R_x} = T_m + a(T_m - T_e) \tag{2}$$

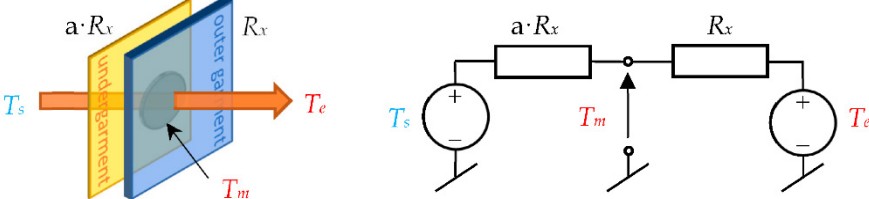

**Figure 8.** The thermal model used for clothing. The input quantities are given in red, and the output in blue.

### 5.2. Effect of Sensor Construction

The slope observed in the external sensor readings indicates a significant thermal time constant. This secondary problem is caused by the construction of the sensor, which restricts airflow around the measuring component. For the intended use, the sensor must be robust and the construction of the sensor appeared to affect its metrological characteristics.

This problem also affects the clothing sensors, as they are identical to the external sensor; however, it is masked by the longer time constants of the body-clothing system.

The authors attempted to simulate thermal behaviour of the sensor using a single time constant model, as presented in Figure 9. The dependency between the true temperature, $T_{ec}$, and the measured temperature, $T_e$, is given by the time constant $RC$, as in Equation (3). By setting the expected external temperature (scaled Heaviside step) to the input and adjusting the value of $RC$ so that the computed output follows the observed measurements for participant B, the optimisation procedure arrives at $RC = 250$ s. The mean error per sample for the first 18 min is 0.27 °C, which is a satisfactory result.

$$\mathrm{d}T_e = \frac{T_{ec} - T_e}{RC}\mathrm{d}t \tag{3}$$

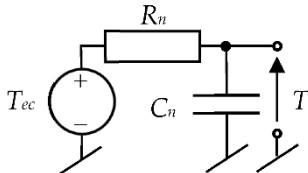

**Figure 9.** Thermal model of the sensor.

### 5.3. Complete Thermal Model

By combining the results from Sections 5.1 and 5.2, a complete model encompassing clothing and sensors can be drawn, as presented in Figure 10. Its parameters can be obtained by an optimisation procedure that requires the value of $T_s$, obtained from the model, to follow the curve of the reference sensor temperature. Unfortunately, this approach results in poor conformance of the model to the reference measurements. The inappropriateness of the constructed model is also highlighted by the fact that relaxing the optimisation constraints and separating the $RC$ values of the clothing and external sensors results in negative clothing sensor capacitance values.

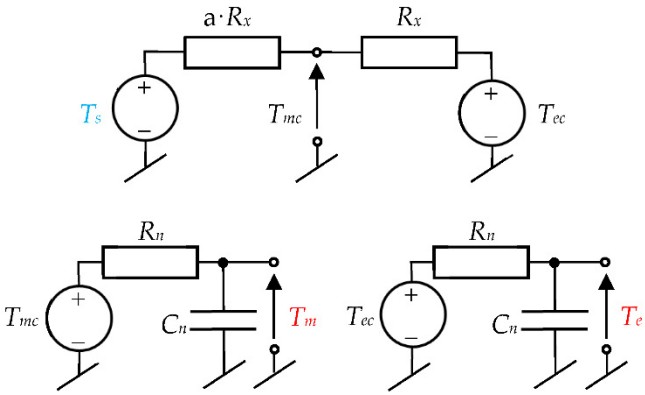

**Figure 10.** Combined thermal models of the sensors and clothing—initial approach. The input quantities are given in red, and the output in blue.

The authors argued that this behaviour may have been caused by the thermal capacitance of the clothing being overlooked in the experimental model. Furthermore, it may not be justifiable to assume that the clothing and external sensors share identical RC values: although they have identical constructions, their working conditions are significantly different. The external sensor relies solely on convective heat transfer, while the internal sensor is also influenced by thermal conduction. The final thermal model is presented in

Figure 11. The change of temperature across thermal capacitance $C_x$, denoted $T_{cx}$, can be computed from Equation (4), and the skin temperature from Equation (5).

$$dT_{cx} = \frac{T_{ec} + T_{mc} - 2T_{cx}}{R_x C_x} dt \tag{4}$$

$$T_s = a(T_{mc} - T_{cx}) + T_{mc} \tag{5}$$

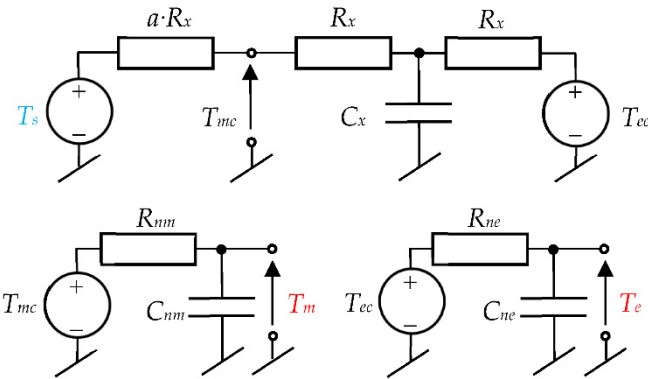

**Figure 11.** Final thermal models of the sensors and clothing. The input quantities are given in red, and the output in blue.

With this model, an optimisation routine was used to elicit the thermal resistance and capacitance values; however, $R_{ne}$ and $C_{ne}$ were set according to the results obtained in Section 4, i.e., so that the *RC* constant equalled 250 s. Note that the values of $R_x$, $R_{nm}$ and $C_{nm}$ were set to be the same for each participant, while the coefficient *a* was allowed to be different. The Generalised Reduced Gradient [27] algorithm was employed, with the loss function defined as in Equation (6), where $T_s$ is the computed and $T_r$ measured (reference) skin temperature. The loss function was computed for each participant for 1000 s from the moment of entering the freezer chamber.

$$L = \sum_{t=1}^{1000} [T_s(t) - T_r(t)]^4 \tag{6}$$

The resulting values are presented in Table 4. The table reports values of $a/(a + 2)$, as they allow the influence of the underwear to be easily compared, being the ratio of resistance between skin and the sensor to the total resistance between skin and external environment.

**Table 4.** Model parameters.

| Parameter | Value |
|---|---|
| $R_x \cdot C_x$ | 300 s |
| $R_{nm} \cdot C_{nm}$ | 31 s |
| $R_{ne} \cdot C_{ne}$ | 249 s |
| $a/(a + 2)$—participant A | 0.17 |
| $a/(a + 2)$—participant B | 0.2 |
| $a/(a + 2)$—participant C | 0.375 |

*5.4. Model Evaluation*

The temperature curves resulting from the model are presented in Figure 12. The overall accuracy is good; in most cases, the difference between the computed and reference temperatures does not exceed 0.5 °C. The greatest discrepancies (reaching 1.67 °C at one point) are observed for participant A; these are caused by fluctuations in the external temperature measurements that do not properly reflect the true behaviour of the external

environment (cf. Figure 7 and discussion in Section 4). Nevertheless, even in this case, the discrepancy does not last long, and given the inherent inertia of thermal phenomena, should not adversely affect the thermal comfort of the user.

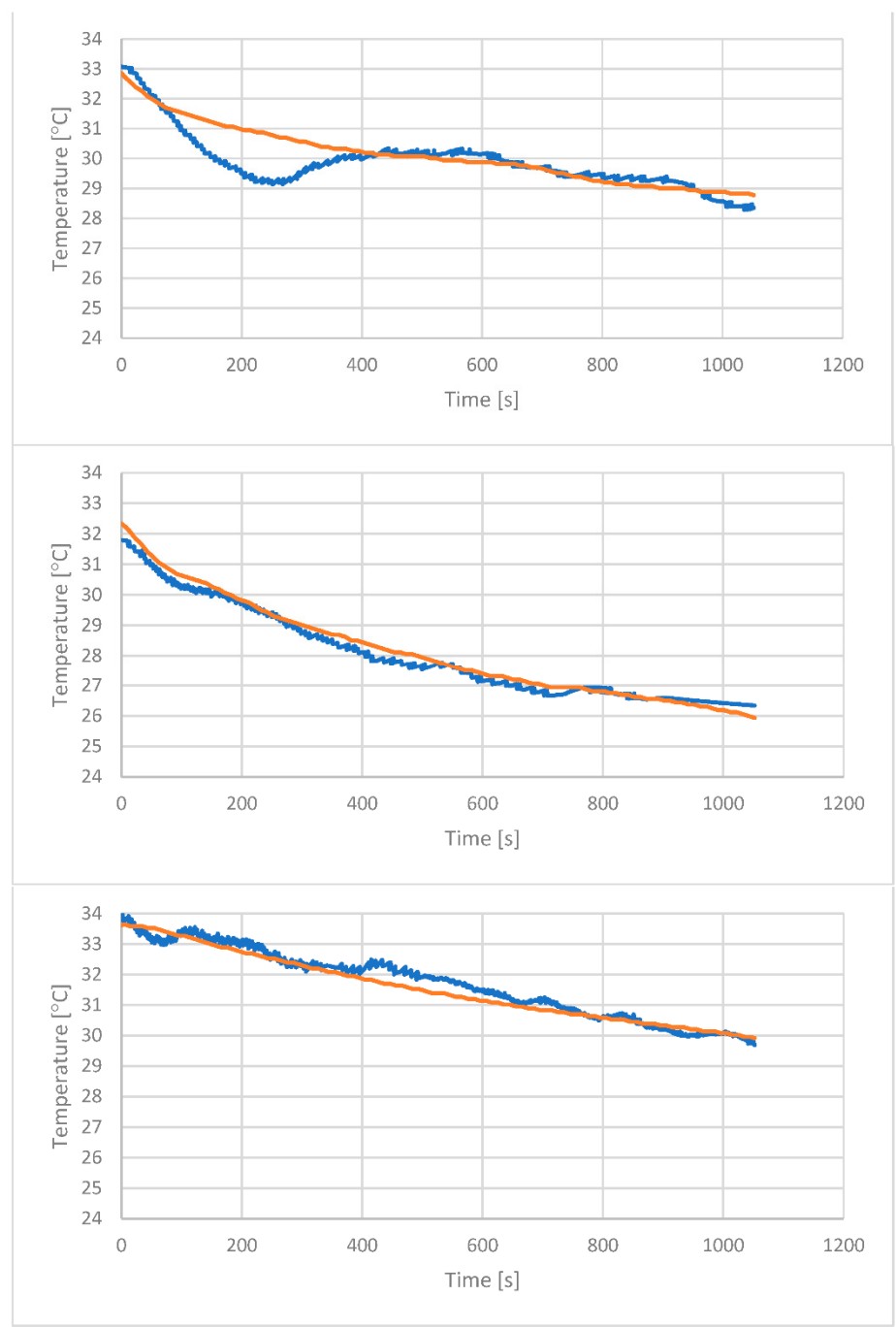

**Figure 12.** Reference (orange) and computed (blue) skin temperatures. From top: participant A, B and C. Note different time scale than in Figure 7.

A particular point of interest is the model error for the moment when the users declared their thermal sensation changed from "neutral" to "too cold". These results are gathered in Table 5. Again, the results are satisfactory. N.B. different temperatures at which discomfort was observed and the much longer time to reach discomfort for participant A. The first observation confirms the fact that thermal comfort is subjective and the control algorithm must adapt to the user's preferences. The reasons for the second phenomenon can only be conjectured. One of the possibilities is that participant A had significantly larger muscle

mass than participants B and C, resulting in slower cooling (heat loss is proportional to the body surface area, while metabolic heat production is related to the body volume, making strongly built persons better adapted to cold climate). Notwithstanding these variations, the model provides satisfactory precision.

**Table 5.** Model results for sensation change moment.

| Participant | Time to Reach | Reference Temperature | Model Temperature | Difference |
|---|---|---|---|---|
| A | 2070 s | 27.27 °C | 27.61 °C | −0.34 °C |
| B | 372 s | 28.62 °C | 28.33 °C | 0.29 °C |
| C | 485 s | 31.52 °C | 31.96 °C | −0.44 °C |

## 6. Conclusions

Our results and model evaluation show that it is possible to create a thermal model that, when used for input data processing, allows undergarment temperature to be converted to skin temperature. Of particular importance is the fact that the model can account for different upper body undergarments, and that it still holds in the presence of transients. Furthermore, the model relies only on the data already measured by the system and uses a single parameter to differentiate the type of undergarment. However, its further evaluation is necessary, in more complex scenarios.

**Author Contributions:** Conceptualisation, W.T., R.K. and M.K.; data curation, W.T. and R.K.; formal analysis, W.T.; investigation, W.T., R.K. and S.W.; methodology, W.T., R.K. and A.D.; software, M.K.; validation, W.T. and R.K.; visualisation, W.T.; writing—original draft, W.T.; writing—review and editing, R.K., M.K. and A.D. All authors have read and agreed to the published version of the manuscript.

**Funding:** This work was supported by the Project "Personalized Protective Thermally Active clothiNg" from The National Centre for Research and Development under Grant POIR.04.01.04-00-0070/18. This work has been completed while the 4th author was a Doctoral Candidate in the Interdisciplinary Doctoral School at the Lodz University of Technology, Poland.

**Informed Consent Statement:** Informed consent was obtained from all subjects involved in the study.

**Data Availability Statement:** Data from the experiments are not publicly available due to foreseen commercial exploitation of the project's results.

**Conflicts of Interest:** The authors declare no conflict of interest. The funders had no role in the design of the study; in the collection, analyses, or interpretation of data; in the writing of the manuscript, or in the decision to publish the results.

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
