# Peer review of "A Thermal Model for Processing Data from Undergarment Sensors in Automatic Control of Actively Heated Clothing"

_energies, doi:10.3390/en15010169_

Round 1

Reviewer 1 Report

attached

Reviewer 2 Report

The work reports a model that can be used to relate undergarment temperature with skin temperature based on the data measured by the imbedded  sensing system. The work is interesting, but I think the work can be more attractive if some points are addressed as follows.

  1. The size of the imbedded sensors should be introduced, which may affect the tactile comfort when human body are contacting with the garment during wearing. Also, the stiffness of the sensors can be a factor that affect the tactile comfort. The authors should pay some attention to this issue.
  2. The state of the art for the current smart thermal management clothing should be reviewed in the introduction part. Some examples are given for your reference. (Review of clothing for thermal management with advanced materials. Cellulose, 2019, 26: 6415-6448; Hierarchically structured and scalable artificial muscles for smart textiles. ACS Applied Materials & Interfaces, 2021, 13: 54386–54395; Novel design of integrated thermal functional garment for primary dysmenorrhea relief. Textile Research Journal, 2020, 90: 1002-1023).
  3. In Figure 7, it is better to indicate the means of each curve in the figure directly.

Reviewer 3 Report

Dear authors,

kindly go through the comments attached

Round 2

Reviewer 1 Report

The authors responded to all suggestion and overall manuscript is now in good form to be accept as it is.

Reviewer 3 Report

well answeered